# A New Analytical Model for Deflection of Concrete Beams Reinforced by BFRP Bars and Steel Fibres under Cyclic Loading

**DOI:** 10.3390/polym14091797

**Published:** 2022-04-28

**Authors:** Haitang Zhu, Zongze Li, Qun Chen, Shengzhao Cheng, Chuanchuan Li, Xiangming Zhou

**Affiliations:** 1School of Civil Engineering, Henan University of Engineering, Zhengzhou 451191, China; htzhu@zzu.edu.cn; 2School of Water Conservancy Engineering, Zhengzhou University, Zhengzhou 450001, China; chenqun9958@163.com (Q.C.); lichuanchuan1004@126.com (C.L.); 3China Construction Seventh Engineering Division, Co., Ltd., Zhengzhou 450004, China; chengshengzhao@aliyun.com; 4Installation Engineering Co., Ltd. of CSCEC 7th Division, Zhengzhou 450004, China; 5Department of Civil & Environmental Engineering, Brunel University London, Uxbridge UB8 3PH, UK; xiangming.zhou@brunel.ac.uk

**Keywords:** cyclic loading, deflection, BFRP-RC beams, steel fiber, analytical model

## Abstract

Basalt-fiber-reinforced plastic-bars-reinforced concrete beams (i.e., BFRP-RC beams) usually possess significant deformations compared to reinforced concrete beams due to the FRP bars having a lower Young’s modulus. This paper investigates the effects of adding steel fibers into BFRP-RC beams to reduce their deflection. Ten BFRP-RC beams were prepared and tested to failure via four-point bending under cyclic loading. The experimental variables investigated include steel-fiber volume fraction and shape, BFRP reinforcement ratio, and concrete strength. The influences of steel fibers on ultimate moment capacity, service load moment, and deformation of the BFRP-RC beams were investigated. The results reveal that steel fibers significantly improved the ultimate moment capacity and service load moment of the BFRP-RC beams. The deflection and residual deflection of the BFRP-RC beams reinforced with 1.5% by volume steel fibers were 48.18% and 30.36% lower than their counterpart of the BFRP-RC beams without fibers. Under the same load, the deflection of the beams increased by 11% after the first stage of three loading and unloading cycles, while the deflection increased by only 8% after three unloading and reloading cycles in the second and third stages. Finally, a new analytical model for the deflection of the BFRP-RC beams with steel fibers under cyclic loading was established and validated by the experiment results from this study. The new model yielded better results than current models in the literature.

## 1. Introduction

The corrosion of steel bars in RC structures shortens the service life of RC structures and significantly increases maintenance costs. Over the past decades, fibre-reinforced polymer (FRP) bars have been used extensively in the construction industry as a new type of reinforcement that replaces steel bars to solve corrosion problems [1,2]. Compared with steel bars, FRP bars are corrosion-free, magnetically transparent, and lighter but with higher tensile strength [3]. These important features enable FRP-RC structures to withstand various complex and corrosive environments with desirable performances. Based on the raw materials used for manufacturing FRPs, FRPs are divided into four categories, including basalt-fiber-reinforced plastic (BFRP), aramid-fiber-reinforced plastic (AFRP), glass-fiber-reinforced plastic (GFRP) and carbon-fiber-reinforced plastic (CFRP). CFRP has a higher Young’s modulus and tensile strength than any other FRP, but its high price hinders its wider applications in construction. Although GFRP and AFRP are less expensive, their alkali resistance is poor, leading to a large degree of strength degradation when reinforcing concrete with alkalinity of pH 12~13. In this regard, BFRP is now used in more applications in construction due to its relatively low cost, good thermal resistance, and excellent freeze/thaw resistance [4]. At the same time, the bond performance between BFRP bars and concrete is also better than that between GFRP bars and concrete. More importantly, BFRP is made of volcanic basalt, which is widely recognized as a type of green construction material with high sustainability credentials; therefore, BFRP has greater application prospects [5]. However, the Young’s modulus of FRP bars is lower than that of steel bars, which leads to FRP-RC structures usually possessing larger crack widths and deflection than RC structures [6,7,8]. In addition, FRP-RC structures are prone to brittle failure because FRP bars are usually brittle while steel bars are ductile.

To solve the above problems, researchers have proposed various ways to improve the structural performance of FRP-RC beams, which include composite reinforcement composed of steel core and FRP to reinforce RC beams [9], increasing the transverse reinforcement ratio for FRP-RC beams [10], grouting FRP bars in corrugated sleeves to reinforce RC beams [11] and adding fibers as additional reinforcement into FRP-RC beams [12]. Previous studies have demonstrated that fibers can improve ductility and strain-hardening of cementitious composite mortars and grouts [13]. Adding discrete fibers into a concrete matrix is the most effective way to improve the serviceability performances of FRP-RC structures. More importantly, the bridging effect of fibers leads to the pseudo-ductile behavior of FRP-RC structures [14,15,16,17,18,19,20,21,22,23,24,25,26,27,28,29]. Chellapandian et al. [14] investigated the effects of adding macro-synthetic fibers into concrete on the cracking, stiffness, and deformability of GFRP-RC beams. They concluded that the fibers improved the post-cracking behavior of GFRP-RC beams with their deformation largely enhanced by adding only 1% by volume of steel fibers, which also transformed the GFRP-RC beams from brittle flexure–shear failure to ductile flexural failure with higher pseudo-ductility. Filipe et al. [15] found that the use of fibers enhanced the stiffness of FRP-RC members and helped to reduce crack spacing and width. The same findings were also obtained by other researchers [16,17]. Ibrahim and Eswari [18] investigated the strength and ductility of GFRP = laminated RC beams incorporated with various amounts of discrete steel fibers. Their study revealed that incorporating steel fibers can effectively improve the strength and ductility of FRP-RC beams. Issa et al. [19] concluded that polypropylene fibers, glass fibers, and steel fibers all improved the ductility of FRP-RC beams, in particular, in the case of steel fibers, which increased the ductility of the beams by 277.8%. Short discrete fibers not only improved the tensile properties of concrete but also improved the shear capacity of concrete beams [20,21,22,23]. Zhu et al. [24] studied the effects of partially steel fibers reinforced concrete (SFRC) on the flexural behavior of FRP-RC beams. The results suggest that compared with full section SFRC beams, partially reinforced SFRC beams cannot provide a better performance, and steel fibers helped to reduce the deflection of the FRP-RC structures. Similar findings were also reported by other researchers. In summary, previous studies have systematically studied the flexural behaviors of FRC beams strengthened with FRP bars under static loading, including crack behaviors, ductility, deflection, and ultimate moment capacity.

However, for practical purposes, RC beams always bear cyclic loading rather than static loading [30]. The flexural behaviors of RC beams under static loading are different from those under cyclic loading, so it is imperative to investigate the effects of the deterioration of concrete and FRP bars on the flexural performances of FRP-RC beams under cyclic loading. Zhu et al. [27] studied the influence of steel fiber on the bearing capacity of the BFRP-RC beams, and the results showed that steel fiber can help to increase the ultimate compressive strain of concrete so as to increase the bearing capacity of the BFRP reinforced concrete beams, and the calculation method of flexural capacity of the BFRP bars and steel-fiber-reinforced concrete beams was established. Li et al. [28,29] analyzed the influence of steel fibers on crack width and ductility of the BFRP-RC beams; the results showed that steel fibers were beneficial in reducing crack width and increasing the ductility of the BFRP-RC beams. The calculation method of maximum crack width and the evaluation method of ductility were put forward. However, the research on the dRC beams with steel fibers under cyclic load is still rare in literature.

The purpose of this research is to study the volume fraction and type of steel fibers on the deformation and flexural behavior of the BFRP-RC beams. In addition, the effects of concrete strength and BFRP reinforcement ratio on the deformation and flexural behavior of the BFRP-RC beams were also investigated. Ten beams were prepared and loaded via four-point bending under cyclic loading until failure, which included nine beams reinforced by both BFRP bars and steel fibers and one BFRP-RC beam without steel fibers as a reference case. The responses of the beams under cyclic loading were compared and analyzed from the aspects of failure mode, ultimate moment capacity, service load moment, load-deflection relation, envelop curves, residual deflection, and stiffness degradation. In addition, a new analytical model for the deformation of the BFRP-RC beams with steel fibers was proposed. Compared with other models, the newly proposed model in this paper accords better to experimental results.

## 2. Materials and Methods

### 2.1. Material Properties

In this study, three types of steel fibers were used as reinforcement for concrete, which were differentiated in the number of hook-ends, length, diameter, and tensile strength, but with the same aspect ratio (i.e., fiber length-to-diameter ratio). Based on the number of hook-ends, the three types of fibers were named 3D, 4D, and 5D, respectively. Three kinds of steel fibers were produced by Bekaert company in Shanghai, China. The steel fibers having only one hook-end were denoted as 3D fibers, as shown in Figure 1a; those with one and a half hook-ends were named 4D fibers, as shown in Figure 1b; and those with two steel fiber hook-ends were called 5D fibers, as shown in Figure 1c. The physical properties and dimensions of the three types of steel fibers are shown in Table 1. It can be seen that the 4D and 5D fibers both had a length of 60 mm, which was longer than that, i.e., 35 mm, of 3D fibers. In addition, 5D fibers had the highest tensile strength among the three types of fibers.

The BFRP bars were produced by Jiangsu lvcaigu new material technology development Co., Ltd. The BFRP bars used as the reinforcement had a diameter of 12 or 14 mm. Their tensile strength and Young’s modulus were measured conforming to Chinese standard GB/T 30022-2013 [31], with the average of the results of five samples taken as the relevant representative value as shown in Table 2.

In this study, two grades of concrete were prepared with the characteristic strength of 30 and 60 MPa, respectively. Mix proportions of the two grades of concrete were designed according to JG/T 472–2015 [32], as shown in Table 3. All beam specimens are labelled as “BN-CN-VN-SN”, where “BN” represents the BFRP reinforcement ratio in percentage, “CN” refers to concrete grade, “VN” refers to steel-fiber volume fraction in percentage and “SN” denotes the type of steel fibers used. For example, “B0.56C60V1.0S3” refers to the beam specimen with a BFRP reinforcement ratio of 0.56% (i.e., 2Φ12), concrete strength of 60 MPa, the steel-fiber volume fraction of 1.0%, and 3D steel fibers.

To ensure the slump of both grades of concrete falling between 50 and 70 mm, a polycarboxylate superplasticizer was added to the concrete mixture. Natural gravels with particle size ranging between 5 and 20 mm was used as coarse aggregates when making concrete. The fine aggregate used was river sand, and its particle size was less than 5 mm. The physical properties of aggregate are shown in Table 4. Grade 42.5R ordinary Portland Cement conforming to Chinese standard GB175-2007 [33] was used as the binder in preparing concrete; its physical properties are shown in Table 5. Six 150 × 150 × 150 mm^3^ concrete cubes and six 150 × 150 × 300 mm^3^ concrete prisms were also prepared alongside each beam specimen for measuring compression strength and splitting tensile strength of concrete.

### 2.2. Test Beams

The flexural behaviors of nine BFRP-RC with steel fibers were investigated, together with that of one BFRP-RC beam without steel fibers, which was also investigated as a reference point. All ten beam specimens had the same sizes, i.e., 150 mm wide (b) × 300 mm deep (h) × 2100 mm long (l), and were tested via four-point bending under cyclic loading until failure, as shown in Figure 2. ACI 440 1R-15 [34] stipulates that FRP-RC beams should be designed with concrete crushing failure, and its reinforcement ratio should be greater than 1.4 times of the balanced reinforcement ratio (*ρ_fb_*), which can be obtained via Equation (1).
(1)$$\rho_{fb}=0.85\beta_{1}\frac{f_{c}^{\prime }}{f_{fu}}\frac{E_{f}\varepsilon_{cu}}{E_{f}\varepsilon_{cu}+f_{fu}}$$
where *β*_1_ is a factor, which can be calculated by Equation (2).
(2)$$\beta_{1}=0.85-0.05\times (\frac{f_{c}^{\prime }-28}{7})\geq 0.65$$

The balanced reinforcement ratio of all ten beam specimens was calculated by Equation (1). The design BFRP reinforcement ratios were 0.56% (in the case of 2Φ12 FRP bars), 0.77% (in the case of 2Φ14 FRP bars), and 1.15% (in the case of 3Φ14 FRP bars), respectively. The reinforcement ratios of all beam specimens were higher than 1.4 times the balanced reinforcement ratio. Four levels of steel-fiber volume fractions were investigated, which are 0%, 0.5%, 1.0% and 1.5% respectively. Among them, three types of steel fibers were used as reinforcement when the steel-fiber volume fraction was 1%. Steel stirrups with a diameter of 10 mm and spacing of 75 mm were placed along the whole length of all beam specimens, which ensures that shear failure will not occur. The concrete cover was 25 mm. Figure 2 depicts the details of the beam specimens. Table 6 lists technical details of all beam specimens as well as the actual mechanical properties of concrete making the specimens.

### 2.3. Experiment Setup and Procedure

From Figure 3a, four-point bending tests under cyclic loading mode were carried out on beams using a 2000 kN Hydraulic Press Machine (HPM) together with a load-distribution steel beam. Seven linear voltage differential transformers (LVDTs) were mounted at both supports, midspan, both loading points, and the other two quartile spans of the pure bending zone of a BFRP-RC beam. To capture the strain of the BFRP bars during loading, nine electrical strain gauges with the size of 3 × 2 mm^2^ were attached to the bottom of the BFRP bars, and their distribution on the beam was depicted in Figure 2. Nineteen π-type strain gauges were attached to the top, bottom, and front surfaces of each BFRP-R beam specimen to capture its strain evolution during loading, as shown in Figure 3b. The crack width of concrete at BFRP bar levels was observed by the ZBL-F120 crack width gauge.

The unloading–reloading protocol is depicted in Figure 4. First, the beams were loaded at a rate of 0.5 kN/min until cracking; subsequently, the displacement of the hydraulic pressure head was increased every 6 mm (i.e., actuator) as the loading grade; for example, the displacement of the first loading grade actuator was 6 mm, the displacement of the second loading grade actuator was 12 mm, and so on. At each loading grade, the loading–unloading cycles were done three times until the test beam failed.

## 3. Results and Discussion

This section presents the experimental results of the nine BFRP-RC beams with steel fibers and the one BFRP-RC beam without steel fibers in terms of failure mode, cracking load, service load moment, ultimate moment capacity, cracking moment, load-deflection evolution, envelope curve, residual deflection, and stiffness degradation. The cracking moment (*M_cr_*) of a BFRP-RC beam was defined as the moment when the stress of the BFRP bars quickly increased or the initial concrete cracking occurred. The stabilized moment (*M_s_*) referred to the moment when no new cracks appeared. Table 7 lists the experimental results for all beams tested.

### 3.1. Failure Modes, Service Load Moment, and Ultimate Moment Capacity

Although the BFRP reinforcement ratios of the beam specimens were all greater than 1.4 times the balanced reinforcement ratio as recommended by ACI 440.1R-15, the beam specimens tested in this research exhibited two distinguished failure modes, which are concrete crushing and BFRP bar rupturing. Figure 5 depicts the two failure modes. BFRP bar rupturing occurred in specimens B0.56C60V1.0S-3 and B0.77C60V1.0S-3 (see Figure 5a), while all other beam specimens failed by concrete crushing. As can be seen from Table 7, the number of cracks of specimens B0.56C60V1.0S-3 and B0.77C60V1.0S-3 was less than that of other beam specimens, but their crack width was larger. For the beams that failed by BFRP bar rupture, the stiffness of the beams decreased rapidly after cracking, and the deflection increased sharply. Then the bearing capacity decreased suddenly before the ultimate failure, and BFRP bars were broken, which was companied by a loud sound. Beams exhibited no ductility under this failure mode, which shall be avoided in design. A beam that failed by concrete crushing is shown in Figure 5b. It can be seen that under such a failure mode, multiple cracks but with smaller widths occurred, and horizontal cracks were observed at the top of the beam section. As observed, the beam experienced the following cracking process before ultimate failure: first, small horizontal cracks appeared one by one at the top of the beam section; then, the small horizontal cracks gradually connected and formed a crack, which led to the bulge of concrete in the compression zone; finally, the ultimate moment capacity of the beam was reached. Therefore, the FRP-RC beams with concrete crushing exhibited good ductility [19,27].

The maximum crack width of FRP-RC beams under service load moment was larger than that of RC beams due to FRP bars possessing an anticorrosion property. The CSA code [35] recommends that the maximum crack width of FRP-RC beams in outdoor and indoor service environments shall be less than 0.5 mm and 0.7 mm, respectively. Coastal engineering structures, bridges, and other infrastructure, which frequently experience cyclic loading, are usually constructed in an outdoor service environment. Therefore, the service load moment (*M_ser_*) of an FRP-RC beam was defined as the moment when the maximum crack width reached 0.5 mm. Figure 6 illustrates the service load moment (*M_ser_*) and ultimate moment capacity (*M_u_*) of all beams. It can be seen from Figure 6a that service load moment and ultimate moment capacity increases with the BFRP reinforcement ratios, but the influence of the BFRP reinforcement ratio on the ultimate moment capacity of beams with BFRP bars rupture was significantly higher than that of beams with concrete crushing. The reason is that the ultimate moment capacity of the beams with BFRP bars rupture was determined by the BFRP reinforcement ratio, while the ultimate moment capacity of beams with concrete crushing was dictated by concrete performances. Steel fibers made a significant contribution to improving the performance of concrete, including enhancing concrete’s tensile strength, ultimate compressive strain, and bond strength, which is beneficial for improving the service load moment and ultimate moment capacity of the beams. The service load moment and ultimate moment capacity of the beam with 1.5% by volume steel fibers were 103.3% and 14.2%, respectively, higher than their counterparts of those beams without steel fibers, as shown in Figure 6b. Steel fibers significantly improved the serviceability of the beams under cyclic loading. From Figure 6c, the service performance and ultimate moment capacity of the beams increased with the increase in fiber length and the number of fiber hook-ends. Compared with 3D and 4D steel fibers reinforced beams, the load moment and ultimate moment capacity of beams with 5D steel fibers were higher. The concrete strength also significantly affects the flexural performances of the beams. As can be seen from Figure 6d, the service load moment and ultimate moment capacity of the beams with high strength concrete (i.e., Grade 60) were 17.1% and 25.9%, respectively, higher than those with low strength concrete (i.e., Grade 30).

### 3.2. Load-Deflection Curve and Envelope Curve

Load-deflection curves of beams under cyclic loading were commonly used for examining their flexural behaviours, from which the envelope curve, energy dissipation, residual deflection, stiffness, etc., was able to be derived. The envelope curve was the curve connecting the peak load of all cycles in the load-deflection curve of a beam under cyclic loading. The enclosed area in the load-deflection curve after the unloading–reloading cycle represented the energy dissipation of the beam under this unloading–reloading cycle. The residual deflection was defined as the irrecoverable deflection of a beam after the load was unloaded to 0. The load degradation coefficient meant the reduction coefficient of the peak load at the same displacement in different unloading–reloading cycles. Figure 7 presents the load-deflection curves of all beams tested in this research. The red curves, blue curves, and pink curves indicate the first cycle, the second cycle, and the third cycle envelope curves, respectively, of the load-deflection curves. The envelope curve was also an important index for studying the flexural performance of a beam under cyclic loading. From Figure 7, it is obvious that the load-deflection curves of all beams demonstrate the identical characteristics, i.e., all load-deflection curves increased linearly after cracking; the residual deflection increased with unloading–reloading cycles, especially at larger deflection; the peak load and energy consumption of the beam under the same deflection decreased gradually with the increase of loading–unloading cycles. Figure 8 reproduces the first cycle envelope curves for all beams.

#### 3.2.1. Number of Unloading–Reloading Cycles

From Figure 7, it can be found that unloading–reloading cycles at the same stroke displacement significantly reduced the peak load of a beam. However, the peak load reduction rate decreased with the increase of unloading–reloading cycles. According to the experimental results, the average peak load of the second cycle was 3%~12% lower than that of the first cycle, while the average peak load of the third cycle was only 1%~5.38% lower than that of the second cycle. This was due to greater damage to the beams caused by the increase in loading in the first cycle, leading to the increase in crack width and height, the decrease of the effective area of concrete, and hence the reduction of stiffness. The peak load of the second and third unloading–reloading cycles was lower than that of the first unloading–reloading cycle. But the width and height of cracks after the second and the third unloading–reloading cycles were comparable to those after the first unloading–reloading cycle. The decrease in stiffness was only related to the internal damage and bond between concrete and BFRP bars. Therefore, the reduction rate of peak load was decreased with the increase of unloading–reloading cycles. For example, for beam B1.15C60V1.0S3 with a stroke displacement of 6 mm, the peak load degradation coefficient after the second and the third unloading–reloading cycles were 7.12% and 1.08%, respectively.

More importantly, the deflection of all beams increased with the increase of the number of unloading–reloading cycles under the same applied load. Table 8 shows the deflections of the beams at the first cycle and the deflections after three unloading–reloading cycles under the same applied load. It can be seen from Table 8 that after three loading and unloading cycles of the first stage under the same applied load, the deflections of the beams increased by 11% on average, but after three loading and unloading cycles of the second and third stages under the same applied load, the deflections of the beams increased by only 8% on average. The reason was that the skeleton curves of the beams were bilinear; due to the lower elastic modulus of the BFRP bars, the stress of the BFRP bars increased rapidly after concrete cracking, resulting in a large increase of the deflection after three loading and unloading cycles of the first stage. Therefore, the deflections of the beams under cyclic loading can be calculated by the following formula.
(3)$$\Delta^{\prime }=\Delta \times (1+11\%)\times {\left(1+8\%\right)}^{n-1}$$
where Δ*_n_*′ is the deflection of a beam after three cycles under cyclic loading, Δ is the deflection of the beam under static loading, and *n* is the loading grade under cyclic loading (n≥1).

#### 3.2.2. BFRP Reinforcement Ratio

Compared with other variables, the BFRP reinforcement ratio had the greatest influence on both the envelope and the load-displacement curves. The BFRP reinforcement ratio directly affects the failure modes of beams under bending. There was an obvious difference between the load-deflection curves of beams that failed by concrete crushing and those that failed by the rupture of BFRP bars. For the beams failed by BFRP bar ruptures, the crack width and height developed rapidly after cracking, which caused the slope of the load-deflection curves to decrease rapidly. More importantly, the peak load decreased with the increase in deflection. For beams B0.56C60V1.0S3 and B0.77C60V1.0S3, the peak load reached the maximum when the displacement was 36 mm, but the peak load decreased at the displacement of 42 mm, as shown in Figure 8a. Therefore, the beams failed by BFRP bars rupture exhibited poor ductility. The slope of the load-deflection curves of the beams with concrete crushing decreased gently after cracking. Moreover, the energy consumption of the beams that failed by concrete crushing was much higher than that of the beams that failed by BFRP bars rupture. The deflection of beams B0.77C60V1.0S3, B1.15C60V1.0S3, and B1.65C60V1.0S3 was 39.57%, 43.78%, and 62.95%, respectively, lower than that of beam B0.56C60V1.0S3 at the applied load of 110 kN.

#### 3.2.3. Steel Fiber Volume Fraction and Shape

Remarkably, the envelope curve of the BFRP-RC beams with steel fibers was different from that of the BFRP-RC beam without steel fibers, as shown in Figure 8b,c. The slope of the first cycle envelope curve of the BFRP-RC beams with steel fibers decreased slowly after cracking, and the first cycle envelope curves were approximately trilinear. However, the first cycle envelope curve of the BFRP-RC beam without steel fibers is approximately bilinear. From Figure 8b, it can be seen that the envelope curve of beam B1.15C60 had the same features as the BFRP-RC beams with steel fibers before cracking, but the bridging effect of steel fibers after cracking limited the further development of crack width and height, which resulted in the stiffness of the beam decreased slowly and the slope of the first cycle envelope curve reduced slowly. The steel-fiber volume fraction of beam B1.15C60V1.0S3 was 1.5% which was higher than that of beam B1.15C60, and the deflection of the former was reduced by 48.18% at 110 kN applied load compared with the latter. The increase in the number of fiber hook-ends was beneficial to improve the stiffness and thus reduce the deflection of the beam. As shown in Figure 8c, when the number of steel fiber hook-ends increased from 1 to 2 (i.e., from 3D to 5D), the deflection of the beam reduced by 11.56% at the applied load of 110 kN.

#### 3.2.4. Concrete Strength

From Figure 8d, the envelope curves of beams with high strength concrete and low strength concrete exhibited the same features during both the elastic growth stage and the rising plastic stage. The load was shared by BFRP bars and concrete matrix before cracking, and the envelope curves demonstrated a linear increase manner until crack occurred. The stress and strain of the BFRP bars increase linearly. After cracks appeared, the first cycle envelope curve increased with the increase of deflection until failure, but the slope of the first cycle envelope curve decreased. This was because the width and height of the crack increased with the increase of deflection, resulting in the reduction of the stiffness of beams. High strength concrete had higher Young’s modulus and tensile strength than low strength concrete, resulting in that the stiffness of beam B1.15C60V1.0S3 was larger than that of beam B1.15C30V1.0S3. Moreover, the area surrounded by the load-deflection curve of beam B1.15C60V1.0S3 was higher than that of beam B1.15C30V1.0S3, suggesting that increasing concrete strength is beneficial for increasing energy consumption, improving stiffness, and reducing deflection of the beam. Compared with beam B1.15C30V1.0S3 with low strength concrete grade 30, the energy dissipation of beam B1.15C60V1.0S3 with high strength concrete grade 60 increased by 2.67% before failure, but the deflection of the beam at 110 kN was reduced by 17.54%. Therefore, the deflection of FRP-RC beams can be effectively reduced by increasing concrete strength [21,22,23].

### 3.3. Residual Deflection

The residual deflection was defined as the irrecoverable deflection of a beam after the load was unloaded to 0 [36]. Figure 9 presents the load-residual deflection curves of beams. From Figure 9, it can be found that the residual deflection of all beams increased with the increase of the applied load and the number of unloading–reloading cycles under the same deflection. Moreover, the influence of the number of unloading–reloading cycles on the residual deflection became more significant under higher load. For beam B1.15C60V1.0S3, the residual deflection after the third unloading–reloading cycle was only 5.15% higher than that of the first loading when the deflection was 6 mm, while the residual deflection after the third unloading–reloading cycle was nearly 10% higher than that of the first loading when the deflection increased to 42 mm. The residual deflection of other beams demonstrated a similar trend. The reason was that the stiffness of a beam was larger at the initial stage of loading, the unloading–reloading cycle had a little cumulative effect on the internal damage of the beam, but the stiffness of the beam degraded rapidly at the later stage of loading, the damage accumulation of concrete and BFRP bars increased with the increase of a number of unloading–reloading cycles, resulting in larger residual deflection.

Figure 10 illustrates the load-residual deflection curves of the beams under the first unloading–reloading cycle. The influences of the four variables on the load-residual deflection curves of the beams are elaborated in Figure 10. The BFRP reinforcement ratio had the greatest influence on the load-residual deflection curves. The stress growth rate of beams with a high BFRP reinforcement ratio was lower than that of beams with a low BFRP reinforcement ratio after cracking. Therefore, beams with a low reinforcement ratio had a larger residual deflection. The residual deflection of B0.77C60V1.0S3, B1.15C60V1.0S3, and B1.65C60V1.0S3 under the 110 kN load was 40.31%, 62.61%, and 76.13%, respectively, lower than that of B0.56C60V1.0S3.

Secondly, concrete strength and steel-fiber volume fraction also had significant effects on the load-residual deflection curves. From Figure 10a,b, the residual strength of the beam decreased with the increase of concrete strength and steel-fiber volume fraction. Compared with beam B1.15C30V1.0S3 with low-strength concrete grade 30, the residual deflection of beam B1.15C60V1.0S3 with high-strength concrete grade 60 at 110 kN reduced by increased by 5.56%. The increase of steel-fiber volume fraction increased the bridging action between concrete and steel fibers, which hindered the development of concrete cracking in the tensile zone, and then enhanced the stiffness of the beams, therefore reducing their residual deflection [36]. Compared with beam B1.15C60 without steel fibers, the residual deflection of beam B1.15C60V1.5S3 with a steel-fiber volume fraction of 1.5% at 110 kN applied load was reduced by 30.36%.

Finally, the steel fiber shape had the least influence on the load-residual deflection curves, as shown in Figure 10d. 3D, 4D, and 5D steel fibers all had the same fiber aspect ratio but different lengths, numbers of hook-ends, and tensile strength. The results indicated that all three types of steel fibers had a bond-slip failure, and no steel fibers were broken during testing. Longer fibers and more hook-ends are both beneficial for improving the bond between concrete and steel fibers, but the strength of steel fibers had little influence on the bond between concrete and fibers. Therefore, the residual deflection of the beams with 5D steel fibers was lower than that of beams with 3D steel fibers. However, the influence of steel fiber shape on the residual deflection was less obvious than the other three variables.

### 3.4. Stiffness Degradation

The deflection of the beams increased with the increase of the unloading–reloading cycles under the same applied load, which was called stiffness degradation. According to JGJ/T 101-2015 [37], the stiffness of a beam is expressed by secant stiffness *K_ij_*, which can be calculated by the following equation.
(4)$$K_{ij}=\frac{\left|+F_{ij}\right|+\left|-F_{ij}^{\prime }\right|}{\left|+\Delta_{ij}\right|+\left|-\Delta_{ij}^{\prime }\right|}$$
where *F_ij_* represents the peak load of the *j_th_* cycle under a displacement of *i_th_*; Δ*_ij_* represents the largest displacement of the *j_th_* cycle under a displacement of *i_th_*; *F_ij_*’ represents the minimum load of the *j_th_* cycle under a displacement of *i_th_*; Δ*_ij_*’ represents the residual deflection of the *j_th_* cycle under a displacement of *i_th_*; *j* is the number of cycles under a displacement of *i_th_*, where *j* is less than or equal to 3 in this study.

As the loading mode was cyclic in this research, *F_ij_*″ = 0, Δ*_ij_*′ = 0. Therefore, the secant stiffness *K_ij_* can be simplified as the following equation:(5)$$K_{ij}=\frac{\left|+F_{ij}\right|}{\left|+\Delta_{ij}\right|}$$

Figure 11 depicts the stiffness–displacement curves of all beams. The stiffness of the beams decreased with the increase of displacement, and the stiffness degradation rate decreased with the increase of displacement. In particular, the stiffness degradation rate was the highest from the initiation of cracking to an actuator displacement of 6 mm. The stiffness of the beams remained unchanged before cracking. After cracking to an actuator displacement of 6 mm, the crack width and height increased rapidly, and the effective section of a beam decreased accordingly, leading to a higher rate of stiffness degradation. When the actuator displacement reached 6 mm, the crack height of a beam changed little, and the stiffness degradation was small. Noticeably, increasing the number of unloading–reloading cycles decreased the stiffness under the same deflection, but the stiffness degradation rate of beams decreased. The stiffness of the beams in the second cycle was 4.00% lower than in the first cycle under the same deflection, and their stiffness in the third cycle was 1.59% lower than in the second cycle. The main reason for this was that after the first cycle, new cracks appeared, and old cracks further developed, leading to rapid stiffness degradation. The peak load of the second cycle decreased under the same displacement, and no new cracks appeared, which had little effect on the stiffness of the beams.

Figure 12 illustrates stiffness–displacement curves of the beams in the first cycle under different variables. The stiffness had increased with the increase of the BFRP reinforcement ratio, but the stiffness degradation rate decreased. After cracking, the restraint force of the beams with a higher reinforcement ratio on crack width expansion was higher than that of the beams with a lower reinforcement ratio, so the stiffness degradation rate of beams with a higher reinforcement ratio was reduced. The increase of steel-fiber volume fraction and number of hook-ends helped to enhance the stiffness. From Figure 12b,c, it can be found that steel-fiber volume fraction and the number of hook-ends had a significant effect on the stiffness–displacement curve of the beams in the early stage of loading, but the effect became less significant in the later stage. Increasing volume fraction and number of hook-ends of steel fibers was beneficial for improving the tensile strength of concrete, and the random distribution of fibers helped to hinder the further development of cracks, thus reducing the deflection of the beams in the early stage of loading. The effects of fibers on deflection were reduced at the later stage of loading because most steel fibers in the tensile zone were pulled out at the ultimate failure. 5D steel fibers had higher tensile strength and more hook-ends than the 3D and 4D steel fibers. Therefore, the bond strength between concrete and fibers was higher than other steel fibers, which made the stiffness of beam B1.15C60V1.0S5 higher than that of beams B1.15C60V1.0S3 and B1.15C60V1.0S4. The effect of concrete strength on beam stiffness–displacement curves are depicted in Figure 12d. Increasing concrete strength can increase the stiffness of the beams, but it has little effect on the stiffness degradation rate.

## 4. Experimental Results versus Model Prediction

FRP-RC beams usually possess larger deflection than RC beams due to the FRP bars having a lower Young’s modulus. In this regard, the serviceability limit states usually control the structural design of FRP-RC beams. Controlling the deformation of FRP-RC beams under cyclic loading is particularly important for design. At present, most studies and design codes use the effective moment of inertia method to evaluate the deflection of FRP-RC beams under static loading, which is also used in this paper to predict and evaluate the deflections of the BFRP-RC beams with steel fibers under cyclic loading. Results from various analytical models/empirical equations were compared with experimental results from this research, through which the analytical model/empirical equations were evaluated for their appropriateness for calculating the deflection of the BFRP-RC beams with steel fibers under cyclic loading. Table 7 summarizes the comparisons between the experimental and theoretical results of the deflection of beams tested at a crack width of 0.5 mm.

### 4.1. Theoretical Calculation of Deflection of FRP-RC Beams

To simplify the analysis, the following assumptions were taken when evaluating the deflection of FRP-RC beams.


(1)A beam section is homogeneous before concrete cracking, and the contribution of the BFRP bars to the total moment of inertia of a beam section is neglected. Therefore, the total moment of inertia (*I_g_*) can be obtained by the following equation.
(6)$$I_{g}=\frac{bh^{3}}{12}$$(2)After a crack is initiated in concrete, the contribution of the concrete in the tension zone is neglected. Therefore, the moment of inertia (*I_cr_*) of the cracked beam section can be obtained by the following equation.
(7)$$I_{cr}=\frac{b}{3}d^{3}k^{3}+n_{f}A_{f}d^{2}{\left(1-k\right)}^{2}$$
(8)$$k=\sqrt{2\rho_{f}n_{f}+{\left(\rho_{f}n_{f}\right)}^{2}}-\rho_{f}n_{f}$$
(9)$$n_{f}=\frac{E_{f}}{E_{c}}$$
where *d* is the effective depth of the beam section, *k* is the ratio of the depth of the neutral axis to the depth of reinforcement bars, *n_f_* is the ratio of Young’s modulus of FRP bars to the modulus of elasticity of concrete, *E_c_* is the Young’s modulus of concrete, *E_f_* is the Young’s modulus of FRP bars, and *ρ_f_* is the FRP reinforcement ratio.


Currently, there are various calculation models for the effective moment of inertia (*I_e_*) of an FRP-RC beam section. Bischoff [38,39] recommended that the effective moment of inertia (*I_e,bischoff_*) of an FRP-RC beam section can be obtained by Equation (10).
(10)$$I_{e,bischoff}=\frac{I_{cr}\;}{1-\left(1-\frac{I_{cr}}{I_{g}}\right){\left(\frac{M_{cr}}{M_{a}}\right)}^{2}}$$
where *M_cr_* is the cracking moment, and *M_a_* is the applied moment.

According to Benmokrane et al. [40], the effective moment of inertia (*I_e,benmokrane_*) of an FRP-RC beam section can be evaluated by Equation (11).
(11)$$I_{e,benmokrane}={\left(\frac{M_{cr}}{M_{a}}\right)}^{3}\frac{I_{g}}{7}+0.84\left[1-{\left(\frac{M_{cr}}{M_{a}}\right)}^{3}\right]I_{cr}\leq I_{g}$$

Alsayed et al. [41] proposed that the effective moment of inertia (*I_e,alsayed_*) of an FRP-RC beam section can be evaluated by Equation (12).
(12)$$\begin{array}{lll} I_{e,alsayed}=\left(1.4-\frac{2}{15}\left(\frac{M_{a}}{M_{cr}}\right)\right)I_{cr} & for & 1<\frac{M_{a}}{M_{cr}}<3\\ I_{e,alsayed}=I_{cr} & for & 3<\frac{M_{a}}{M_{cr}} \end{array}$$

Canadian ISIS [42] code recommends that the effective moment of inertia (IISIS) of an FRP-RC beam section can be evaluated by Equation (13).
(13)$$I_{e}=\frac{I_{g}I_{cr}\;}{I_{cr}+\left[1-0.5{\left(\frac{M_{cr}}{M_{a}}\right)}^{2}\right](I_{g}-I_{cr})}$$

Combined with classical beam theory and the effective moment of inertia method, the mid-span deflection (Δ) of an FRP-RC beam can be obtained by Equation (14).
(14)$$\Delta =\frac{pl_{a}}{48E_{c}I_{e}}(3l_{o}^{2}-4l_{a}^{2})$$
where *p* is the applied load, *l_a_* is the shear span, and *l_o_* is the clear span.

Using the various effective moment of inertia formulas in literature and design codes summarized above, combined with the deflection calculation method from the classical beam theory, the deflection of a BFRP-RC beam with steel fibers at 0.5 mm crack width can be obtained. However, none of the above analytical models for the effective moment of inertia considers the positive contribution of steel fibers to the moment of inertia of the section of a beam strengthened with BFRP bars. Indeed, for a BFRP-RC beam with steel fibers, the contribution of steel fibers in the concrete tensile zone cannot be neglected, because steel fibers reinforced concrete can bear large tensile stress after concrete cracking.

### 4.2. A New Model for the Deflection of the BFRP-RC Beams with Steel Fibers

As elaborated above, when calculating the deflections of the BFRP-RC beams with steel fibers, the effect of steel fibers on the deflection should be considered. However, the influences of steel fibers on the deflection of a BFRP-RC beam with steel fibers should be considered after concrete cracking [19]. The distance from the center of the mass of a fiber to the neutral axis of the beam cannot be calculated, resulting in the inability to obtain its area and moment of inertia. Rather, some scholars believe that the steel fibers in a beam section can be taken as a whole, which can obtain its area and moment of inertia [19]. The distribution and orientation of steel fibers dictate the concrete’s performance, especially at the post-cracking stage. Generally, the distribution of steel fibers is described by the non-uniformity coefficient *η**_v_*, while the orientation is by the orientation coefficient *η*_0_. Zhu [19] recommends that the total area of steel fibers in an SFRC beam is obtained by the following equations:(15)$$A_{sf}=\eta_{0}\eta_{v}bh\rho_{sf}=\eta bh\rho_{sf}$$
where *η* is the steel fiber effective coefficient (in this research, *η* was taken as 0.16).

Figure 13a depicts the gross section and transformed an uncracked section of a beam. Since the area moments of the compression and tension zones of the beam are equal, Equations (16) and (17) can be derived as follows:(16)$$\frac{bx_{0}^{2}}{2}+\frac{(n_{sf}-1)bx_{0}^{2}A_{sf}}{2h}=\frac{b{\left(h-x_{0}^{}\right)}^{2}}{2}+(n_{sf}-1)A_{sf}\frac{{\left(h-x_{0}\right)}^{2}}{2h}+(n_{f}-1)A_{f}(d-x_{0})$$(17)$$x_{0}=\frac{\frac{bh_{}^{2}}{2}+(n_{f}-1)A_{f}d+\frac{(n_{sf}-1)A_{sf}h}{2h}}{bh+(n_{f}-1)A_{f}+(n_{sf}-1)A_{sf}}$$(18)$$n_{sf}=\frac{E_{sf}}{E_{c}}$$

The moment of inertia of the gross section (*I_g_*) of a BFRP-RC beam with steel fibers is calculated by the following equation.
(19)$$I_{g}=\frac{b}{3}\left[x_{0}^{3}+{\left(h-x_{0}\right)}^{3}\right]+(n_{f}-1)A_{f}{\left(d-x_{0}\right)}^{2}+\frac{(n_{sf}-1)A_{sf}}{3h}\left[x_{0}{}^{3}+{\left(h-x_{0}\right)}^{3}\right]$$

Figure 13b describes the cracked and transformed cracked sections of a beam. Since the area moments of the compression and tension zones of the beam are equal, Equations (20) and (21) can be derived as follows:(20)$$\frac{bx_{cr}^{2}}{2}+\frac{(n_{sf}-1)bx_{cr}^{2}A_{sf}}{2h}=n_{sf}A_{sf}(d-x_{cr})+\frac{n_{sf}{\left(h-x_{cr}\right)}^{2}A_{sf}}{2h}$$(21)$$x_{cr}=\frac{-(n_{sf}A_{sf}+n_{f}A_{f})+\sqrt{{\left(n_{sf}A_{sf}+n_{f}A_{f}\right)}^{2}+2(b-\frac{A_{sf}}{h})(\frac{n_{sf}}{2}hA_{sf}+n_{f}A_{f}d)}}{b-\frac{A_{sf}}{h}}$$

The moment of inertia of the cracked section (*I_cr_*) of a BFRP-RC beam with steel fibers is calculated by the following equation.
(22)$$I_{cr}=\frac{b}{3}x_{cr}^{3}+n_{f}A_{f}{\left(d-x_{cr}\right)}^{2}+\frac{n_{sf}A_{sf}}{3h}{\left(h-x_{cr}\right)}^{3}$$

After the moment of inertia of the gross and cracked section of a BFRP-RC beam with steel fibers was obtained, the deflection of the beam can be calculated by introducing Equations (13) and (14).

According to the loading regime adopted in this research, the deformation of the beam can be checked according to the static loading before the stroke of the actuator reached 6 mm, but the cyclic loading effect shall be considered after the actuator’s stroke reached 6 mm. Table 9 lists the deflections of all beams investigated in this research when the stroke of the actuator reached 6 mm from experiment and calculation from various models in literature and design codes, as well as the new analytical model established in this study. From Table 9, the calculated deflections from the Bischoff, Benmokrane, Alsayed, and Canadian ISIS models are 13%~49% higher than the experimental value. While the average deflections calculated by the proposed analytical model in this paper are only 9% higher than the experimental ones, and the coefficient of variation is only 0.22, which is lower than the coefficient of variation of any other existing model investigated. Therefore, the analytical model proposed in this paper is more reliable and accurate for evaluating the deflection of the BFRP-RC beams with steel fibers.

According to the loading regime adopted in this study, the cyclic loading effect shall be considered after the stroke of the actuator reached 6 mm. In sum, the deflection of a beam under a certain load can be calculated by using Equations (13), (14), (19) and (22). By introducing the calculated results into Equation (1), the deflection of the beam under cyclic loading can be calculated. Table 10 compares the calculated deflection and the actual deflection of the beams after three loading and unloading cycles. From Table 10, it can be seen that the ratio between the calculated value from the model to the counterpart from the experiment was 0.99, and the coefficient of variation was 0.16, suggesting that the analytical model proposed in this paper can accurately evaluate the deflection of the BFRP-RC beams with steel fibers under cyclic loading.

## 5. Conclusions

The main purpose of this paper was to quantify the influences of short steel fibers on the flexural behaviors of the BFRP-RC beams. Ten beams, including nine BFRP-RC beams with steel fibers and one BFRP-RC beam without steel fibers, were tested via four-point bending under cyclic loading. To accurately calculate the deflection of the BFRP-RC beams under serviceability limit states, a modified analytical model for deflection of the BFRP-RC beams with steel fibers under cyclic loading was proposed and compared with the available deflection calculation models for FRP-RC beams without steel fibers. The following main conclusions can be drawn from the results of this research:The service load moment of the BFRP-RC beams with 1.5% by volume steel fibers was 103.3% higher than that of the beams without fibers, and the deflection and the residual deflection of the beams were reduced by 48.18% and 30.36% at the applied load of 100kN. Moreover, increasing the steel-fiber volume fraction can significantly enhance the stiffness of the BFRP-RC beams after cracking.Increasing the number of unloading–reloading cycles reduced the peak load and increased the residual deflection of the BFRP-RC beams under the same deflection. The deflection of the beams increased by 11% after the first stage of three loading and unloading cycles, while the deflection increased by only 8% after three unloading and reloading cycles in the second and third stages of loading.The BFRP reinforcement ratio had the greatest influence on the load-deflection curves, load-residual deflection curves, and stiffness–displacement curves of the BFRP-RC beams. Higher-strength concrete was beneficial in improving the stiffness of the beams and reducing their deflection. A higher BFRP reinforcement ratio was beneficial to improving the serviceability of the BFRP-RC beams, which is the controlling limit state for the structural design of the BFRP-RC beams with steel fibers.Combined with the influences of cyclic loading on the deflection, a new analytical method for evaluating the deflection of the BFRP-RC beams with steel fibers under cyclic loading was proposed in this research, which gives better results than any other available model in literature and design codes when compared with experimental results.

## Figures and Tables

**Figure 1 polymers-14-01797-f001:**
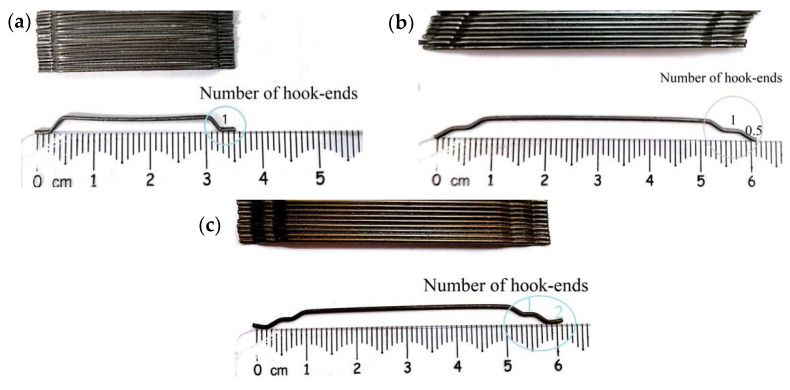
Three types of steel fibers: (**a**) 3D; (**b**) 4D; (**c**) 5D.

**Figure 2 polymers-14-01797-f002:**
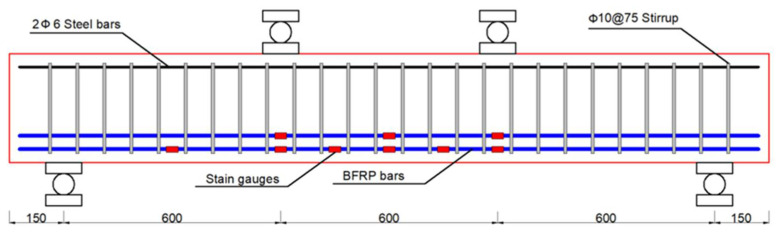
Specimen details (all dimensions in millimeters).

**Figure 3 polymers-14-01797-f003:**
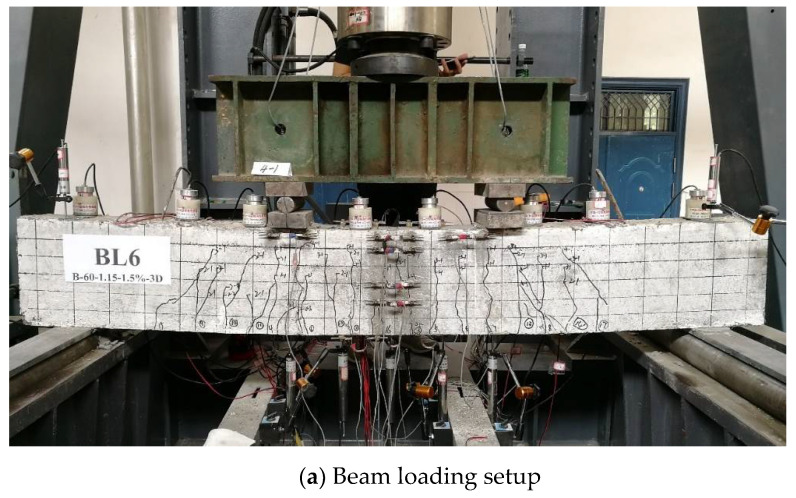
Test setup (dimensions in millimeters).

**Figure 4 polymers-14-01797-f004:**
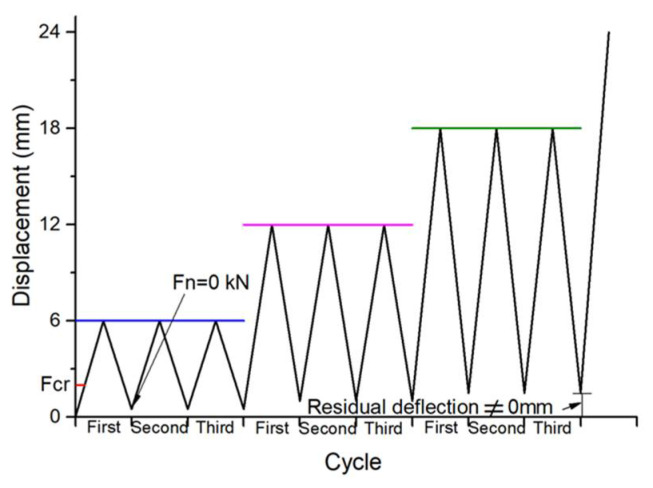
Unloading–reloading process. Note: *F_cr_* is the cracking load; *F_n_* is the load applied to a beam.

**Figure 5 polymers-14-01797-f005:**
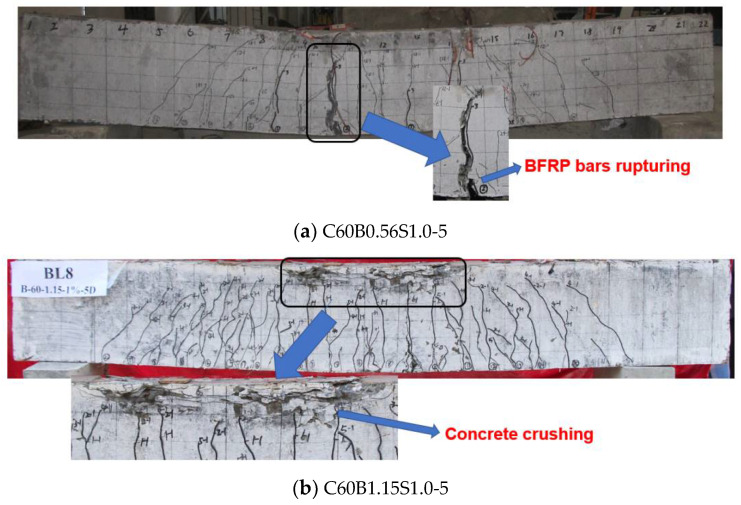
Failure modes of beams: (**a**) BFRP bars rupture; (**b**) concrete crushing.

**Figure 6 polymers-14-01797-f006:**
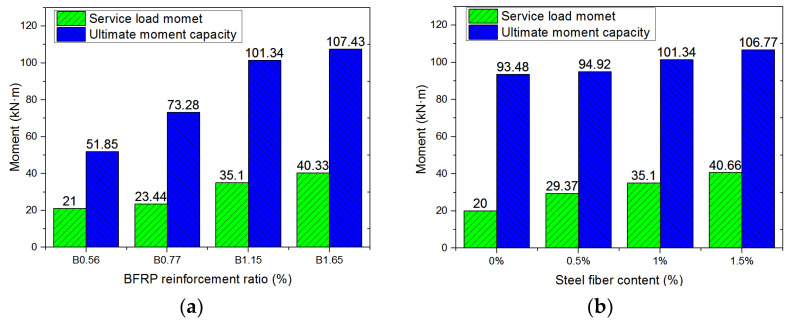
Service load moment and ultimate moment capacity of beams with respect to (**a**) BFRP reinforcement ratio; (**b**) steel fiber content; (**c**) steel fiber shape; (**d**) concrete strength.

**Figure 7 polymers-14-01797-f007:**
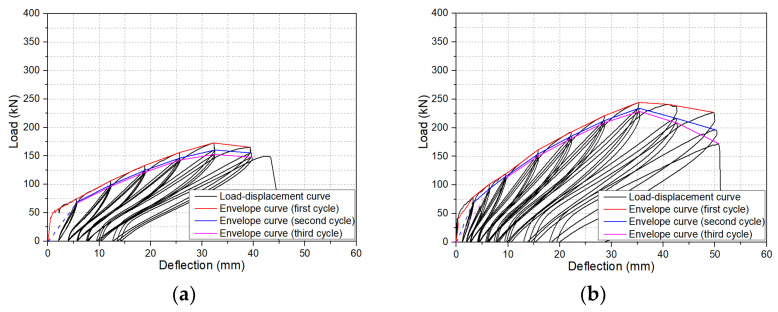
Load-deflection curves for all beams tested: (**a**) B0.56C60V1.0S3; (**b**) B0.77C60V1.0S3; (**c**) B1.15C60V1.0S3; (**d**) B1.65C60V1.0S3; (**e**) B1.15C60; (**f**) B1.15C60V0.5S3; (**g**) B1.15C60V1.5S3; (**h**) B1.15C60V1.0S4; (**i**) B1.15C60V1.0S5; (**j**) B1.15C30V1.0S3.

**Figure 8 polymers-14-01797-f008:**
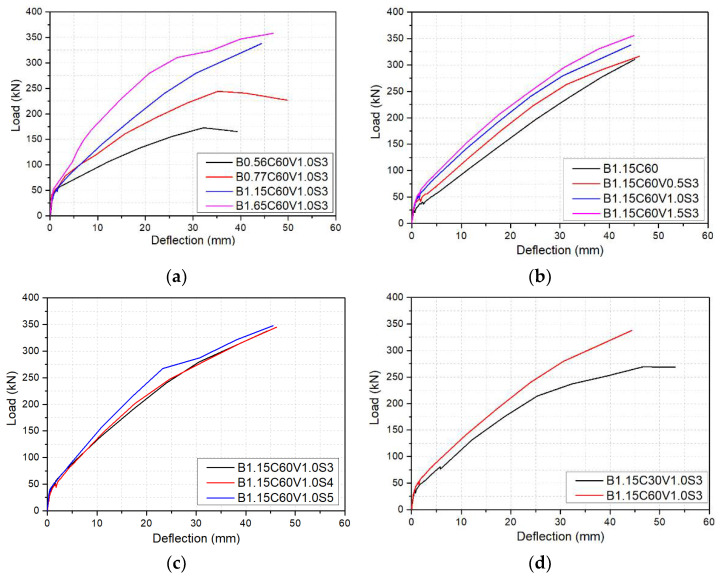
Envelope curve (first cycle) for beams with different: (**a**) BFRP reinforcement ratio; (**b**) steel-fiber volume fraction; (**c**) steel fiber shape; and (**d**) concrete strength.

**Figure 9 polymers-14-01797-f009:**
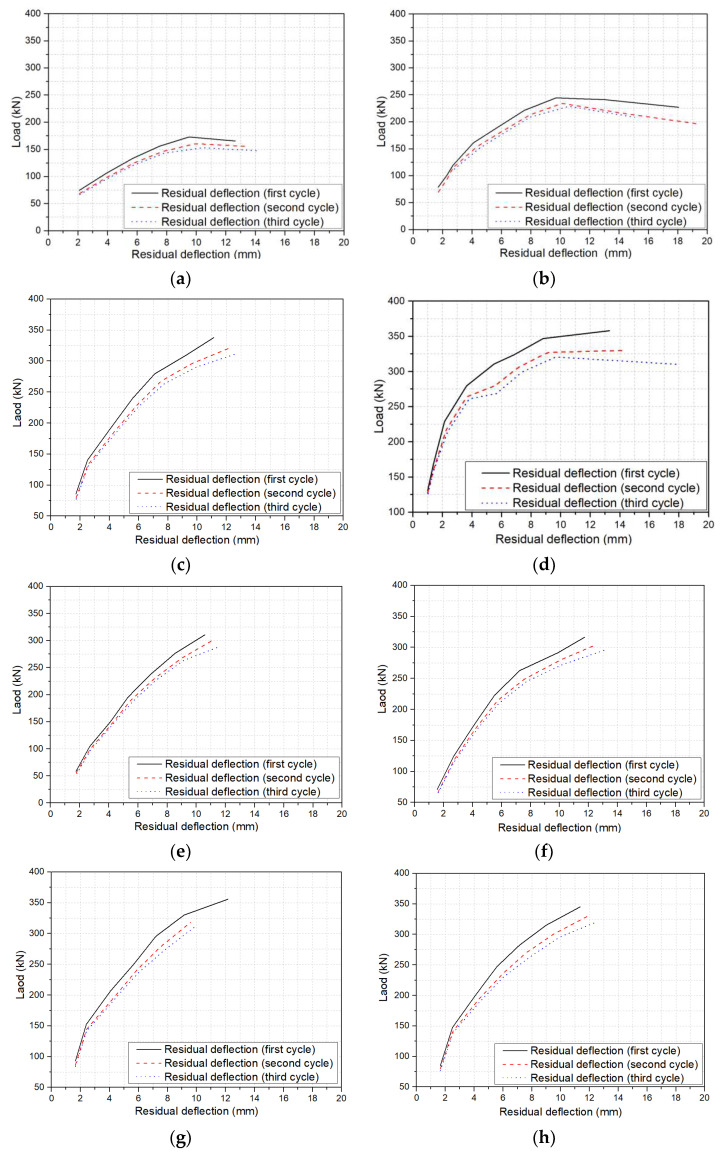
Load-residual deflection curves for beams: (**a**) B0.56C60V1.0S3; (**b**) B0.77C60V1.0S3; (**c**) B1.15C60V1.0S3; (**d**) B1.65C60V1.0S3; (**e**) B1.15C60; (**f**) B1.15C60V0.5S3; (**g**) B1.15C60V1.5S3; (**h**) B1.15C60V1.0S4; (**i**) B1.15C60V1.0S5; (**j**) B1.15C30V1.0S3.

**Figure 10 polymers-14-01797-f010:**
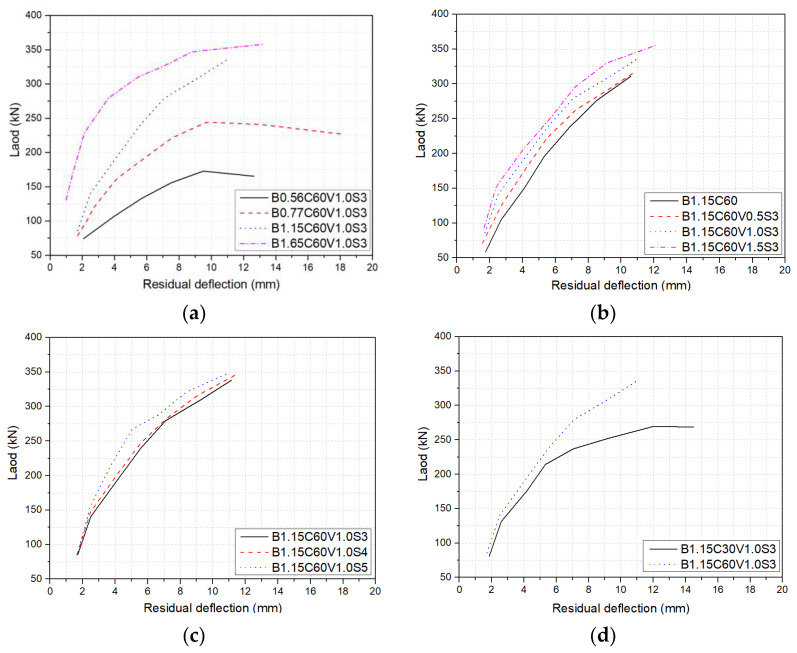
Load-residual deflection (first cycle) for beams with different: (**a**) BFRP reinforcement ratio; (**b**) steel fiber content; (**c**) steel fiber shape; (**d**) concrete strength.

**Figure 11 polymers-14-01797-f011:**
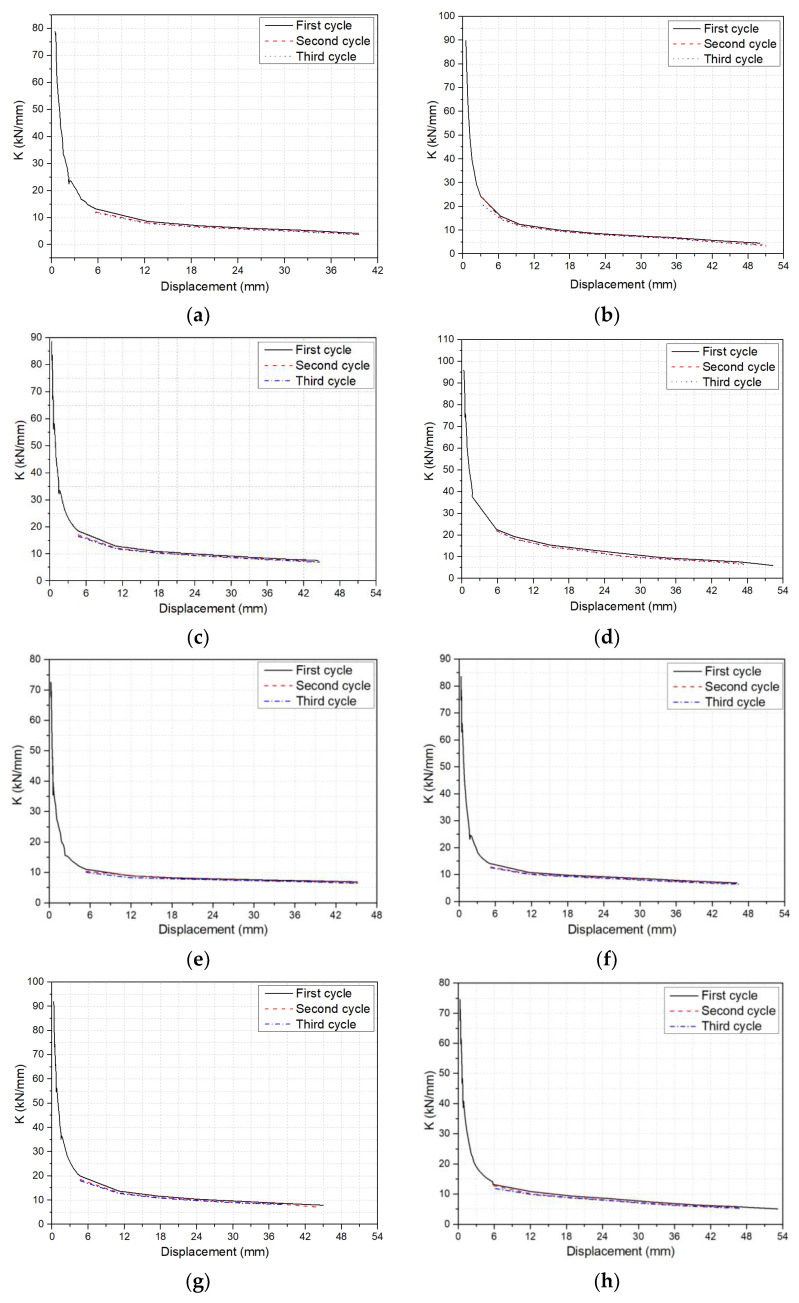
Stiffness–displacement curves: (**a**) B0.56C60V1.0S3; (**b**) B0.77C60V1.0S3; (**c**) B1.15C60V1.0S3; (**d**) B1.65C60V1.0S3; (**e**) B1.15C60; (**f**) B1.15C60V0.5S3; (**g**) B1.15C60V1.5S3; (**h**) B1.15C30V1.0S3; (**i**) B1.15C60V1.0S4; (**j**) B1.15C60V1.0S5.

**Figure 12 polymers-14-01797-f012:**
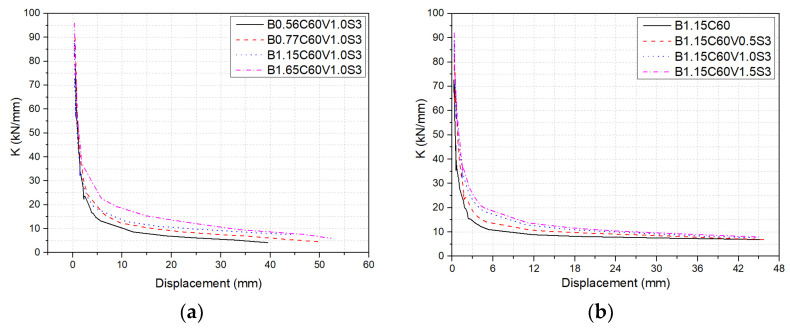
Stiffness–displacement curves of beams at the first loading–unloading cycle with respect to (**a**) reinforcement ratio; (**b**) steel-fiber volume fraction; (**c**) steel fiber shape; (**d**) concrete strength.

**Figure 13 polymers-14-01797-f013:**
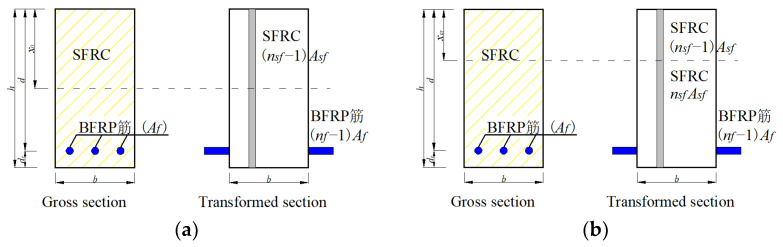
Sectional parameters of the gross and cracked sections: (**a**) gross section; (**b**) cracked section. “筋” = steel.

**Table 1 polymers-14-01797-t001:** Physical properties and dimensions of steel fibers used in this research.

Types	*l_sf_* (mm)	*d_sf_* (mm)	*l_sf_/d_sf_*	*f_t,sf_* (MPa)	*E_sf_* (GPa)	Number of Hook-Ends
3D	35	0.55	65	1345	200	1
4D	60	0.90	65	1600	200	1.5
5D	60	0.90	65	2300	200	2

Note: *l_sf_* is the length of steel fibers; *d_sf_* is the diameter of steel fibers; *f_t,sf_* is the tensile strength of steel fibers; *E_sf_* is the Young’s modulus of steel fibers.

**Table 2 polymers-14-01797-t002:** Tensile properties of the BFRP bars.

Types	Diameter (mm)	Tensile Strength (*f_fu_*) (MPa)	Young Modulus (*E_f_*) (GPa)	Yield Strength
BFRP	12	1080	47.0	—
BFRP	14	1060	46.5	—

**Table 3 polymers-14-01797-t003:** Concrete mixtures (in kg/m^3^) of the specimens.

Beams	Water	Cement	Sand	Steel Fiber	Coarse Aggregate	Polycarboxylate Superplasticizer
B0.56C60V1.0S3	172	521.2	669.3	78.5 (3D)	1013.5	5.212
B0.77C60V1.0S3	172	521.2	669.3	78.5 (3D)	1013.5	5.212
B1.15C60V1.0S3	172	521.2	669.3	78.5 (3D)	1013.5	5.212
B1.65C60V1.0S3	172	521.2	669.3	78.5 (3D)	1013.5	5.212
B1.15C60	172	521.2	648.6	—	1058.2	2.606
B1.15C60V0.5S3	172	521.2	658.9	39.3 (3D)	1035.9	4.170
B1.15C60V1.5S3	172	521.2	679.6	117.8 (3D)	991.1	7.297
B1.15C60V1.0S4	172	521.2	669.3	78.5 (4D)	1013.5	5.212
B1.15C60V1.0S5	172	521.2	669.3	78.5 (5D)	1013.5	5.212
B1.15C30V1.0S3	215	330.8	706.2	78.5 (3D)	1124.0	0

**Table 4 polymers-14-01797-t004:** Physical properties of aggregate.

Aggregate	Specific Gravity	Water Absorption	Fineness Modulus	Free Moisture Content	Graded Zone
Fine aggregate	2.60	1.01%	2.78	2%	II
Coarse aggregate	2.74	0.30%	7.5	NIL	NIL

**Table 5 polymers-14-01797-t005:** Physical properties of the used cement.

Compressive Strength/MPa		Flexural Strength /MPa		Setting Time/min		Specific Surface Area m^2^/kg
3 d	28 d	3 d	28 d	Initial Setting Time	Final Setting Time	
27.8	46.8	5.6	8.5	122	232	345

**Table 6 polymers-14-01797-t006:** Technical details and actual mechanical properties of concrete for all specimens.

Beams	*ρ_f_* (%)	*ρ_sf_* (%)	Steel Fiber Shapes	*f_cu,k_* (MPa)	Actual Physical Properties of Concrete			
					*f_cu_* (MPa)	*f_t_* (MPa)	*f_c_*′ (MPa)	*E_c_* (GPa)
B0.56C60V1.0S3	0.56	1.0	3D	60	60.16	5.59	48.13	41.30
B0.77C60V1.0S3	0.77	1.0	3D	60	74.99	5.70	52.45	42.70
B1.15C60V1.0S3	1.15	1.0	3D	60	81.47	6.60	65.18	42.40
B1.65C60V1.0S3	1.65	1.0	3D	60	76.47	5.84	61.18	42.40
B1.15C60	1.15	0	—	60	74.54	3.56	59.63	41.62
B1.15C60V0.5S3	1.15	0.5	3D	60	69.00	4.88	51.75	41.00
B1.15C60V1.5S3	1.15	1.5	3D	60	81.47	5.17	65.18	42.23
B1.15C60V1.0S4	1.15	1.0	4D	60	83.89	5.83	62.92	42.38
B1.15C60V1.0S5	1.15	1.0	5D	60	79.14	5.51	63.31	43.02
B1.15C30V1.0S3	1.15	1.0	3D	30	44.00	3.36	34.00	35.00

**Table 7 polymers-14-01797-t007:** Experimental results for all beams.

Beams	Failure Modes	*M_cr_* (kN·m)	*M_s_* (kN·m)	*M_u_* (kN·m)	Δ*_max_* (mm)	*ω*_100_ kN (mm)	Number of Cracks
B0.56C60V1.0S3	BFRP bars rupture	13.50	21.07	51.85	32.23	0.72	7
B0.77C60V1.0S3	BFRP bars rupture	14.10	23.40	73.28	35.23	0.70	8
B1.15C60V1.0S3	Concrete crushing	14.25	25.74	101.34	44.32	0.37	10
B1.65C60V1.0S3	Concrete crushing	15.00	28.23	101.43	46.83	0.35	10
B1.15C60	Concrete crushing	9.30	18.14	93.48	44.31	0.75	7
B1.15C60V0.5S3	Concrete crushing	13.50	21.27	94.92	46.03	0.52	9
B1.15C60V1.5S3	Concrete crushing	16.50	27.93	106.77	44.42	0.33	11
B1.15C60V1.0S4	Concrete crushing	15.00	25.50	103.53	46.04	0.35	10
B1.15C60V1.0S5	Concrete crushing	15.00	26.82	104.37	45.50	0.34	11
B1.15C30V1.0S3	Concrete crushing	9.75	22.90	80.50	46.45	0.50	10

Note: *M_cr_* is the cracking moment; *M_s_* is the stabilized moment; *M_u_* is the ultimate moment capacity of a beam; Δ*_max_* is the deflection when the ultimate moment capacity is reached; *ω*_100_ kN is the crack width of a beam at 100 kN.

**Table 8 polymers-14-01797-t008:** Beam’s deflections at the first cycle and after three loading–unloading cycles under the same applied load.

Beams	Load (kN)	Δ_1_ (mm)	Δ_1*′*_ (mm)	Δ_1_/Δ_1*′*_	Load (kN)	Δ_2_ (mm)	Δ_2′_ (mm)	Δ_2_/Δ_2′_	Load (kN)	Δ_3_ (mm)	Δ_3′_ (mm)	Δ_3_/Δ_3′_
B0.56C60V1.0S3	74.43	5.57	6.50	1.17	106.19	12.20	13.56	1.11	133.17	18.85	20.72	1.10
B0.77C60V1.0S3	78.53	3.59	4.20	1.17	119.00	9.54	10.51	1.10	160.93	15.74	17.39	1.10
B1.15C60V1.0S3	85.80	4.63	5.15	1.11	140.40	10.82	11.75	1.09	191.50	17.27	18.50	1.07
B1.65C60V1.0S3	130.90	5.77	6.60	1.14	169.40	8.76	9.50	1.08	229.00	14.87	16.19	1.09
B1.15C60	60.00	5.33	5.86	1.10	105.32	11.79	12.62	1.07	150.39	18.27	19.11	1.05
B1.15C60V0.5S3	71.00	5.02	5.60	1.12	124.90	11.55	12.28	1.06	175.10	17.97	18.95	1.05
B1.15C60V1.5S3	93.10	4.66	5.15	1.11	152.50	11.00	11.86	1.08	206.20	17.51	18.89	1.08
B1.15C60V1.0S4	85.00	4.88	5.30	1.09	147.40	11.19	11.86	1.06	201.70	17.52	18.70	1.07
B1.15C60V1.0S5	89.40	4.75	5.20	1.09	155.40	10.77	11.45	1.06	212.90	16.85	17.90	1.06
B1.15C30V1.0S3	76.33	5.79	6.32	1.09	130.28	12.18	13.27	1.09	175.49	18.62	20.10	1.08
Average value				1.12				1.08				1.08

Note: Δ*_m_* is the deflection of a beam at the first cycle under the mth loading stage, and Δ*_m_*′ is the deflection of the beam after the third unloading–reloading cycle under the mth loading stage.

**Table 9 polymers-14-01797-t009:** Deflections of the beams from the experiment and calculated from the proposed analytical momodelhen the stroke of the actuator reached 6 mm.

Beams	*F*_1_ (kN)	Δ_1_ (mm)	Δ*_c_* (mm)	Δ*_c_* /Δ_1_	Δ*_Benm._* (mm)	Δ*_Benm._* /Δ_1_	Δ*_Bisch._* (mm)	Δ*_Bisch._* /Δ_1_	Δ*_Alsa_* (mm)	Δ*_Alsa_* /Δ_1_	Δ*_ISIS_* (mm)	Δ*_ISIS_* /Δ_1_
B0.56C60V1.0S3	74.43	5.57	6.6	1.20	9.14	1.64	7.84	1.41	9.92	1.78	9.73	1.75
B0.77C60V1.0S3	78.53	3.59	5.84	1.63	8.02	2.23	6.22	1.73	7.96	2.22	7.79	2.17
B1.15C60V1.0S3	85.80	4.63	5.02	1.08	7.10	1.53	5.11	1.10	6.12	1.32	6.10	1.32
B1.65C60V1.0S3	130.90	5.77	7.58	1.31	9.63	1.67	8.36	1.45	9.16	1.59	8.99	1.56
B1.15C60	60.00	5.33	4.37	0.91	5.03	0.94	3.82	0.72	4.40	0.83	4.42	0.83
B1.15C60V0.5S3	71.00	5.02	4.34	0.86	5.72	1.14	3.74	0.74	4.95	0.99	4.81	0.96
B1.15C60V1.5S3	93.10	4.66	4.91	1.05	7.67	1.65	5.25	1.13	6.55	1.41	6.47	1.39
B1.15C60V1.0S4	85.00	4.88	4.89	1.00	7.05	1.44	4.83	0.99	6.00	1.23	5.94	1.22
B1.15C60V1.0S5	89.40	4.75	5.2	1.09	7.38	1.55	5.27	1.11	6.35	1.34	6.33	1.33
B1.15C30V1.0S3	76.33	5.79	5.22	0.84	6.82	1.18	5.40	0.93	6.39	1.10	6.33	1.09
Average value				1.09		1.50		1.13		1.38		1.36
Coefficient of variation				0.22		0.34		0.30		0.38		0.37

Note: *F*_1_ is the applied load on the beams when the stroke of the actuator reached 6 mm; Δ_1_ is the deflection of the beams when the stroke of the actuator reached 6 mm; Δ*_c_* is the deflection of the beams calculated from the analytical model established in this paper; Δ*_Benm._* is the deflection of the beams calculated using Benmokrane’s model; Δ*_Bisch._* is the deflection of the beams calculated using Bischoff’s model; Δ*_Alsa_* is the deflection of the beams calculated from Alsayed’s model; Δ*_ISIS_* is the deflection of the beams calculated using the Canadian ISIS model.

**Table 10 polymers-14-01797-t010:** Deflections of the beams were obtained from the experiment and calculated from the proposed analytical model after three loading and unloading cycles.

Beams	*F*_1_ (kN)	Δ_1′_ (mm)	Δ*_c_*_1′_ (mm)	Δ_*c*1′_ /Δ_1′_	*F*_2_ (kN)	Δ_2′_ (mm)	Δ_*c*2′_ (mm)	Δ_*c*2′_ /Δ_2*′*_	*F*_3_ (kN)	Δ_3′_ (mm)	Δ_*c*3′_ (mm)	Δ*_c_*_3′_ /Δ_3′_
B0.56C60V1.0S3	74.43	6.50	7.33	1.13	106.19	13.56	12.43	0.92	133.17	20.72	17.35	0.84
B0.77C60V1.0S3	78.53	4.20	6.48	1.54	119.00	10.51	11.68	1.11	160.93	17.39	17.66	1.02
B1.15C60V1.0S3	85.80	5.15	5.57	1.08	140.40	11.75	10.73	0.91	191.50	18.50	16.20	0.88
B1.65C60V1.0S3	130.90	6.60	8.41	1.27	169.40	9.50	11.98	1.26	229.00	16.19	17.81	1.10
B1.15C60	60.00	5.86	4.85	0.83	105.32	12.62	11.06	0.88	150.39	19.11	17.42	0.91
B1.15C60V0.5S3	71.00	5.60	4.82	0.86	124.90	12.28	10.43	0.85	175.10	18.95	16.27	0.86
B1.15C60V1.5S3	93.10	5.15	5.45	1.06	152.50	11.86	10.65	0.90	206.20	18.89	15.98	0.85
B1.15C60V1.0S4	85.00	5.30	5.43	1.02	147.40	11.86	11.29	0.95	201.70	18.70	17.11	0.91
B1.15C60V1.0S5	89.40	5.20	5.77	1.11	155.40	11.45	11.91	1.04	212.90	17.90	18.02	1.01
B1.15C30V1.0S3	76.33	6.32	5.79	0.92	130.28	13.27	10.46	0.79	175.49	20.10	15.40	0.77
Average value	0.99											
Coefficient of variation		0.16										

Note: Δ*_cn_*′ is the deflection of the beams calculated from the analytical model established in this paper after three loading and unloading cycles.

## Data Availability

The data presented in this study are available on request from the corresponding author.

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
