# Peer review of "A New Analytical Model for Deflection of Concrete Beams Reinforced by BFRP Bars and Steel Fibres under Cyclic Loading"

_polymers, 2022, doi:10.3390/polym14091797_

Round 1
Reviewer 1 Report
This paper investigates the effect of BFRP bars and steel fibers on deflection of reinforced concrete beams. The topic is very original, and useful for the profession.
The introduction however is relatively light and needs to be strengthened to better situate the context and need to conduct such work, especially that similar topics have been investigated in the past. For example, the authors are recommended to refer to Mechanics of Advanced Materials and Structures, pp. 1-12, 2021. https://doi.org/10.1080/15376494.2021.1882625 , and also to some MDPI papers such as Environments 9, no. 37, 2022. https://doi.org/10.3390/environments9030037
The materials used and experimental program executed are well detailed.
The discussion of results should be improved, while being more succinct. For example, the section between lines 227-267 is too long, difficult to read, and should be better restructured. Section 3.3 should be supported by relevant references, and English writing improved. The conclusion bullet points section should be made more succinct, and straight to the point.
The reviewer recommends publication.
Reviewer 2 Report
The article investigates the possibility of reducing the deflection of concrete beams reinforced with basalt fiber reinforced plastic bars. The influence of steel fibers on the deflection of the beams was the focus of the study. Ten beams were tested under cyclic flexural four-point loading to evaluate the effect of the investigated parameters, which were fiber volume fraction, fiber configuration, concrete compressive strength and BFRP bars reinforcement ratio. In addition to the experimental work, the article introduces a simplified sectional analysis to evaluate the deflection of the tested beams. The experimental work presented in the article is interesting, the presentation of the results is adequate and the discussion is satisfactory. However, this reviewer believes that considering the following recommendations would enhance the final quality of the article.
- The focus of the research is on the flexural performance of BFRP reinforced fibrous concrete beams under cyclic loading, while none of the closely related researches was explored in the literature review presented in the introduction section. The authors should extend the literature about this point and better review the previously published articles (references 27 to 29). Based on this revision, the authors should clearly highlight the novelty of this article showing the experimental differences between this work and the above mentioned previous works.
- The physical properties and chemical composition of the used cement were not presented in section 2. A table that shows the characteristics of the used cement should be added.
- The authors should also present the grading of the adopted sand and gravel in section 2.
- In section 3, it would be more beneficial for readers if the final retained residual deflections of the nine beams are normalized by those of the reference beam and presented in bar charts. To better show the effect of the investigated parameters on the deflections of the beams, the bar charts of the normalized final deflections can be categorized in four figures just like those presented in Figure 6.
- In section 4, presenting the stress diagrams of the transformed sections depicted in Figure 13 would make it easier for readers to follow the formulation introduced in Equations 16 to 22.
- The effect of concrete compressive strength on the flexural behavior and especially on the retained deflections needs to be better clarified in the third conclusion.
